# Adaptive Newton Method for Empirical Risk Minimization to Statistical Accuracy

**Aryan Mokhtari**[*]
University of Pennsylvania
aryanm@seas.upenn.edu

**Hadi Daneshmand**[*]
ETH Zurich, Switzerland
hadi.daneshmand@inf.ethz.ch

**Aurelien Lucchi**
ETH Zurich, Switzerland
aurelien.lucchi@inf.ethz.ch

**Thomas Hofmann**
ETH Zurich, Switzerland
thomas.hofmann@inf.ethz.ch

**Alejandro Ribeiro**
University of Pennsylvania
aribeiro@seas.upenn.edu

## Abstract

We consider empirical risk minimization for large-scale datasets. We introduce Ada Newton as an adaptive algorithm that uses Newton's method with adaptive sample sizes. The main idea of Ada Newton is to increase the size of the training set by a factor larger than one in a way that the minimization variable for the current training set is in the local neighborhood of the optimal argument of the next training set. This allows to exploit the quadratic convergence property of Newton's method and reach the statistical accuracy of each training set with only one iteration of Newton's method. We show theoretically that we can iteratively increase the sample size while applying single Newton iterations without line search and staying within the statistical accuracy of the regularized empirical risk. In particular, we can double the size of the training set in each iteration when the number of samples is sufficiently large. Numerical experiments on various datasets confirm the possibility of increasing the sample size by factor 2 at each iteration which implies that Ada Newton achieves the statistical accuracy of the full training set with about two passes over the dataset.[1]

## 1 Introduction

A hallmark of empirical risk minimization (ERM) on large datasets is that evaluating descent directions requires a complete pass over the dataset. Since this is undesirable due to the large number of training samples, stochastic optimization algorithms with descent directions estimated from a subset of samples are the method of choice. First order stochastic optimization has a long history [19, 17] but the last decade has seen fundamental progress in developing alternatives with faster convergence. A partial list of this consequential literature includes Nesterov acceleration [16, 2], stochastic averaging gradient [20, 6], variance reduction [10, 26], and dual coordinate methods [23, 24].

When it comes to stochastic second order methods the first challenge is that while *evaluation* of Hessians is as costly as evaluation of gradients, the stochastic *estimation* of Hessians has proven more challenging. This difficulty is addressed by incremental computations in [9] and subsampling in [7] or circumvented altogether in stochastic quasi-Newton methods [21, 12, 13, 11, 14]. Despite this incipient progress it is nonetheless fair to say that the striking success in developing stochastic first order methods is not matched by equal success in the development of stochastic second order methods. This is because even if the problem of estimating a Hessian is solved there are still four challenges left in the implementation of Newton-like methods in ERM:

**(i)** Global convergence of Newton's method requires implementation of a line search subroutine and line searches in ERM require a complete pass over the dataset.

**(ii)** The quadratic convergence advantage of Newton's method manifests close to the optimal solution but there is no point in solving ERM problems beyond their statistical accuracy.

**(iii)** Newton's method works for strongly convex functions but loss functions are not strongly convex for many ERM problems of practical importance.

**(iv)** Newton's method requires inversion of Hessians which is costly in large dimensional ERM.

Because stochastic Newton-like methods can't use line searches [cf. (i)], must work on problems that may be not strongly convex [cf. (iii)], and never operate very close to the optimal solution [cf (ii)], they never experience quadratic convergence. They do improve convergence constants and, if efforts are taken to mitigate the cost of inverting Hessians [cf. (iv)] as in [21, 12, 7, 18] they result in faster convergence. But since they still converge at linear rates they do not enjoy the foremost benefits of Newton's method.

In this paper we attempt to circumvent (i)-(iv) with the Ada Newton algorithm that combines the use of Newton iterations with adaptive sample sizes [5]. Say the total number of available samples is $N$, consider subsets of $n \leq N$ samples, and suppose the statistical accuracy of the ERM associated with $n$ samples is $V_n$ (Section 2). In Ada Newton we add a quadratic regularization term of order $V_n$ to the empirical risk – so that the regularized risk also has statistical accuracy $V_n$ – and assume that for a certain initial sample size $m_0$, the problem has been solved to its statistical accuracy $V_{m_0}$. The sample size is then increased by a factor $\alpha > 1$ to $n = \alpha m_0$. We proceed to perform a single Newton iteration with unit stepsize and prove that the result of this update solves this extended ERM problem to its statistical accuracy (Section 3). This permits a second increase of the sample size by a factor $\alpha$ and a second Newton iteration that is likewise guaranteed to solve the problem to its statistical accuracy. Overall, this permits minimizing the empirical risk in $\alpha/(\alpha - 1)$ passes over the dataset and inverting $\log_\alpha N$ Hessians. Our theoretical results provide a characterization of the values of $\alpha$ that are admissible with respect to different problem parameters (Theorem 1). In particular, we show that asymptotically on the number of samples $n$ and with proper parameter selection we can set $\alpha = 2$ (Proposition 2). In such case we can optimize to within statistical accuracy in about 2 passes over the dataset and after inversion of about $3.32 \log_{10} N$ Hessians. Our numerical experiments verify that $\alpha = 2$ is a valid factor for increasing the size of the training set at each iteration while performing a single Newton iteration for each value of the sample size.

## 2 Empirical risk minimization

We aim to solve ERM problems to their statistical accuracy. To state this problem formally consider an argument $\mathbf{w} \in \mathbb{R}^p$, a random variable $Z$ with realizations $z$ and a convex loss function $f(\mathbf{w}; z)$. We want to find an argument $\mathbf{w}^*$ that minimizes the statistical average loss $L(\mathbf{w}) := \mathbf{E}_Z[f(\mathbf{w}, Z)]$,

$$\mathbf{w}^* := \underset{\mathbf{w}}{\operatorname{argmin}} \, L(\mathbf{w}) = \underset{\mathbf{w}}{\operatorname{argmin}} \, \mathbf{E}_Z[f(\mathbf{w}, Z)]. \tag{1}$$

The loss in (1) can't be evaluated because the distribution of $Z$ is unknown. We have, however, access to a training set $\mathcal{T} = \{z_1, \dots, z_N\}$ containing $N$ independent samples $z_1, \dots, z_N$ that we can use to estimate $L(\mathbf{w})$. We therefore consider a subset $\mathcal{S}_n \subseteq \mathcal{T}$ and settle for minimization of the empirical risk $L_n(\mathbf{w}) := (1/n) \sum_{k=1}^{n} f(\mathbf{w}, z_k)$,

$$\mathbf{w}_n^\dagger := \underset{\mathbf{w}}{\operatorname{argmin}} \, L_n(\mathbf{w}) = \underset{\mathbf{w}}{\operatorname{argmin}} \, \frac{1}{n} \sum_{k=1}^{n} f(\mathbf{w}, z_k), \tag{2}$$

where, without loss of generality, we have assumed $\mathcal{S}_n = \{z_1, \dots, z_n\}$ contains the first $n$ elements of $\mathcal{T}$. The difference between the empirical risk in (2) and the statistical loss in (1) is a fundamental quantities in statistical learning. We assume here that there exists a constant $V_n$, which depends on the number of samples $n$, that upper bounds their difference for all $\mathbf{w}$ with high probability (w.h.p),

$$\sup_{\mathbf{w}} |L(\mathbf{w}) - L_n(\mathbf{w})| \leq V_n, \qquad \text{w.h.p.} \tag{3}$$

That the statement in (3) holds with w.h.p means that there exists a constant $\delta$ such that the inequality holds with probability at least $1 - \delta$. The constant $V_n$ depends on $\delta$ but we keep that dependency

implicit to simplify notation. For subsequent discussions, observe that bounds $V_n$ of order $V_n = O(1/\sqrt{n})$ date back to the seminal work of Vapnik – see e.g., [25, Section 3.4]. Bounds of order $V_n = O(1/n)$ have been derived more recently under stronger regularity conditions that are not uncommon in practice, [1, 8, 3].

An important consequence of (1) is that there is no point in solving (2) to an accuracy higher than $V_n$. Indeed, if we find a variable $\mathbf{w}$ for which $L_n(\mathbf{w}_n) - L_n(\mathbf{w}^\dagger) \leq V_n$ finding a better approximation of $\mathbf{w}^\dagger$ is moot because (3) implies that this is not necessarily a better approximation of the minimizer $\mathbf{w}^*$ of the statistical loss. We say the variable $\mathbf{w}_n$ solves the ERM problem in (2) to within its statistical accuracy. In particular, this implies that adding a regularization of order $V_n$ to (2) yields a problem that is essentially equivalent. We can then consider a quadratic regularizer of the form $cV_n/2\|\mathbf{w}\|^2$ to define the regularized empirical risk $R_n(\mathbf{w}) := L_n(\mathbf{w}) + (cV_n/2)\|\mathbf{w}\|^2$ and the corresponding optimal argument

$$\mathbf{w}_n^* := \underset{\mathbf{w}}{\mathrm{argmin}}\, R_n(\mathbf{w}) = \underset{\mathbf{w}}{\mathrm{argmin}}\, L_n(\mathbf{w}) + \frac{cV_n}{2}\|\mathbf{w}\|^2. \tag{4}$$

Since the regularization in (4) is of order $V_n$ and (3) holds, the difference between $R_n(\mathbf{w}_n^*)$ and $L(\mathbf{w}^*)$ is also of order $V_n$ – this may be not as immediate as it seems; see [22]. Thus, we can say that a variable $\mathbf{w}_n$ satisfying $R_n(\mathbf{w}_n) - R_n(\mathbf{w}_n^*) \leq V_n$ solves the ERM problem to within its statistical accuracy. We accomplish this goal in this paper with the Ada Newton algorithm which we introduce in the following section.

## 3 Ada Newton

To solve (4) suppose the problem has been solved to within its statistical accuracy for a set $\mathcal{S}_m \subset \mathcal{S}_n$ with $m = n/\alpha$ samples where $\alpha > 1$. Therefore, we have found a variable $\mathbf{w}_m$ for which $R_m(\mathbf{w}_m) - R_m(\mathbf{w}_m^*) \leq V_m$. Our goal is to update $\mathbf{w}_m$ using the Newton step in a way that the updated variable $\mathbf{w}_n$ estimates $\mathbf{w}_n^*$ with accuracy $V_n$. To do so compute the gradient of the risk $R_n$ evaluated at $\mathbf{w}_m$

$$\nabla R_n(\mathbf{w}_m) = \frac{1}{n}\sum_{k=1}^{n} \nabla f(\mathbf{w}_m, z_k) + cV_n\mathbf{w}_m, \tag{5}$$

as well as the Hessian $\mathbf{H}_n$ of $R_n$ evaluated at $\mathbf{w}_m$

$$\mathbf{H}_n := \nabla^2 R_n(\mathbf{w}_m) = \frac{1}{n}\sum_{k=1}^{n} \nabla^2 f(\mathbf{w}_m, z_k) + cV_n\mathbf{I}, \tag{6}$$

and update $\mathbf{w}_m$ with the Newton step of the regularized risk $R_n$ to compute

$$\mathbf{w}_n = \mathbf{w}_m - \mathbf{H}_n^{-1}\nabla R_n(\mathbf{w}_m). \tag{7}$$

Note that the stepsize of the Newton update in (7) is 1, which avoids line search algorithms requiring extra computation. The main contribution of this paper is to derive a condition that guarantees that $\mathbf{w}_n$ solves $R_n$ to within its statistical accuracy $V_n$. To do so, we first assume the following conditions are satisfied.

**Assumption 1.** *The loss functions $f(\mathbf{w}, \mathbf{z})$ are convex with respect to $\mathbf{w}$ for all values of $\mathbf{z}$. Moreover, their gradients $\nabla f(\mathbf{w}, \mathbf{z})$ are Lipschitz continuous with constant $M$*

$$\|\nabla f(\mathbf{w}, \mathbf{z}) - \nabla f(\mathbf{w}', \mathbf{z})\| \leq M\|\mathbf{w} - \mathbf{w}'\|, \qquad \textit{for all } \mathbf{z}. \tag{8}$$

**Assumption 2.** *The loss functions $f(\mathbf{w}, \mathbf{z})$ are self-concordant with respect to $\mathbf{w}$ for all $\mathbf{z}$.*

**Assumption 3.** *The difference between the gradients of the empirical loss $L_n$ and the statistical average loss $L$ is bounded by $V_n^{1/2}$ for all $\mathbf{w}$ with high probability,*

$$\sup_{\mathbf{w}} \|\nabla L(\mathbf{w}) - \nabla L_n(\mathbf{w})\| \leq V_n^{1/2}, \qquad \textit{w.h.p.} \tag{9}$$

The conditions in Assumption 1 imply that the average loss $L(\mathbf{w})$ and the empirical loss $L_n(\mathbf{w})$ are convex and their gradients are Lipschitz continuous with constant $M$. Thus, the empirical risk

---
**Algorithm 1** Ada Newton
---
1: **Parameters:** Sample size increase constants $\alpha_0 > 1$ and $0 < \beta < 1$.
2: **Input:** Initial sample size $n = m_0$ and argument $\mathbf{w}_n = \mathbf{w}_{m_0}$ with $\|\nabla R_n(\mathbf{w}_n)\| < (\sqrt{2c})V_n$
3: **while** $n \leq N$ **do** {main loop}
4:     Update argument and index: $\mathbf{w}_m = \mathbf{w}_n$ and $m = n$. Reset factor $\alpha = \alpha_0$ .
5:     **repeat** {sample size backtracking loop}
6:         Increase sample size: $n = \min\{\alpha m, N\}$.
7:         Compute gradient [cf. (5)]: $\nabla R_n(\mathbf{w}_m) = (1/n)\sum_{k=1}^{n} \nabla f(\mathbf{w}_m, z_k) + cV_n \mathbf{w}_m$
8:         Compute Hessian [cf. (6)]: $\mathbf{H}_n = (1/n)\sum_{k=1}^{n} \nabla^2 f(\mathbf{w}_m, z_k) + cV_n \mathbf{I}$
9:         Newton Update [cf. (7)]: $\mathbf{w}_n = \mathbf{w}_m - \mathbf{H}_n^{-1}\nabla R_n(\mathbf{w}_m)$
10:        Compute gradient [cf. (5)]: $\nabla R_n(\mathbf{w}_n) = (1/n)\sum_{k=1}^{n} \nabla f(\mathbf{w}_n, z_k) + cV_n \mathbf{w}_n$
11:        Backtrack sample size increase $\alpha = \beta\alpha$.
12:    **until** $\|\nabla R_n(\mathbf{w}_n)\| < (\sqrt{2c})V_n$
13: **end while**
---

$R_n(\mathbf{w})$ is strongly convex with constant $cV_n$ and its gradients $\nabla R_n(\mathbf{w})$ are Lipschitz continuous with parameter $M + cV_n$. Likewise, the condition in Assumption 2 implies that the average loss $L(\mathbf{w})$, the empirical loss $L_n(\mathbf{w})$, and the empirical risk $R_n(\mathbf{w})$ are also self-concordant. The condition in Assumption 3 says that the gradients of the empirical risk converge to their statistical average at a rate of order $V_n^{1/2}$. If the constant $V_n$ in condition (3) is of order not faster than $O(1/n)$ the condition in Assumption 3 holds if the gradients converge to their statistical average at a rate of order $V_n^{1/2} = O(1/\sqrt{n})$. This is a conservative rate for the law of large numbers.

In the following theorem, given Assumptions 1-3, we state a condition that guarantees the variable $\mathbf{w}_n$ evaluated as in (7) solves $R_n$ to within its statistical accuracy $V_n$.

**Theorem 1.** *Consider the variable $\mathbf{w}_m$ as a $V_m$-optimal solution of the risk $R_m$, i.e., a solution such that $R_m(\mathbf{w}_m) - R_m(\mathbf{w}_m^*) \leq V_m$. Let $n = \alpha m > m$, consider the risk $R_n$ associated with sample set $\mathcal{S}_n \supset \mathcal{S}_m$, and suppose assumptions 1 - 3 hold. If the sample size $n$ is chosen such that*

$$\left(\frac{2(M + cV_m)V_m}{cV_n}\right)^{1/2} + \frac{2(n - m)}{nc^{1/2}} + \frac{\left((2 + \sqrt{2})c^{1/2} + c\|\mathbf{w}^*\|\right)(V_m - V_n)}{(cV_n)^{1/2}} \leq \frac{1}{4} \qquad (10)$$

*and*

$$144\left(V_m + \frac{2(n - m)}{n}(V_{n-m} + V_m) + 2(V_m - V_n) + \frac{c(V_m - V_n)}{2}\|\mathbf{w}^*\|^2\right)^2 \leq V_n \qquad (11)$$

*are satisfied, then the variable $\mathbf{w}_n$, which is the outcome of applying one Newton step on the variable $\mathbf{w}_m$ as in (7), has sub-optimality error $V_n$ with high probability, i.e.,*

$$R_n(\mathbf{w}_n) - R_n(\mathbf{w}_n^*) \leq V_n, \qquad w.h.p. \qquad (12)$$

*Proof.* See Section 4. $\qquad\qquad\qquad\qquad\qquad\qquad\qquad\qquad\qquad\qquad\qquad\qquad\qquad\square$

Theorem 1 states conditions under which we can iteratively increase the sample size while applying single Newton iterations without line search and staying within the statistical accuracy of the regularized empirical risk. The constants in (10) and (11) are not easy to parse but we can understand them qualitatively if we focus on large $m$. This results in a simpler condition that we state next.

**Proposition 2.** *Consider a learning problem in which the statistical accuracy satisfies $V_m \leq \alpha V_n$ for $n = \alpha m$ and $\lim_{n \to \infty} V_n = 0$. If the regularization constant $c$ is chosen so that*

$$\left(\frac{2\alpha M}{c}\right)^{1/2} + \frac{2(\alpha - 1)}{\alpha c^{1/2}} < \frac{1}{4}, \qquad (13)$$

*then, there exists a sample size $\tilde{m}$ such that (10) and (11) are satisfied for all $m > \tilde{m}$ and $n = \alpha m$. In particular, if $\alpha = 2$ we can satisfy (10) and (11) with $c > 16(2\sqrt{M} + 1)^2$.*

*Proof.* That the condition in (11) is satisfied for all $m > \tilde{m}$ follows simply because the left hand side is of order $V_m^2$ and the right hand side is of order $V_n$. To show that the condition in (10) is satisfied for sufficiently large $m$ observe that the third summand in (10) is of order $O((V_m - V_n)/V_n^{1/2})$ and vanishes for large $m$. In the second summand of (10) we make $n = \alpha m$ to obtain the second summand in (13) and in the first summand replace the ratio $V_m/V_n$ by its bound $\alpha$ to obtain the first summand of (13). To conclude the proof just observe that the inequality in (13) is strict. $\square$

The condition $V_m \leq \alpha V_n$ is satisfied if $V_n = 1/n$ and is also satisfied if $V_n = 1/\sqrt{n}$ because $\sqrt{\alpha} < \alpha$. This means that for most ERM problems we can progress geometrically over the sample size and arrive at a solution $\mathbf{w}_N$ that solves the ERM problem $R_N$ to its statistical accuracy $V_N$ as long as (13) is satisfied .

The result in Theorem 1 motivates definition of the Ada Newton algorithm that we summarize in Algorithm 1. The core of the algorithm is in steps 6-9. Step 6 implements an increase in the sample size by a factor $\alpha$ and steps 7-9 implement the Newton iteration in (5)-(7). The required input to the algorithm is an initial sample size $m_0$ and a variable $\mathbf{w}_{m_0}$ that is known to solve the ERM problem with accuracy $V_{m_0}$. Observe that this initial iterate doesn't have to be computed with Newton iterations. The initial problem to be solved contains a moderate number of samples $m_0$, a mild condition number because it is regularized with constant $cV_{m_0}$, and is to be solved to a moderate accuracy $V_{m_0}$ – recall that $V_{m_0}$ is of order $V_{m_0} = O(1/m_0)$ or order $V_{m_0} = O(1/\sqrt{m_0})$ depending on regularity assumptions. Stochastic first order methods excel at solving problems with moderate number of samples $m_0$ and moderate condition to moderate accuracy.

We remark that the conditions in Theorem 1 and Proposition 2 are conceptual but that the constants involved are unknown in practice. In particular, this means that the allowed values of the factor $\alpha$ that controls the growth of the sample size are unknown a priori. We solve this problem in Algorithm 1 by backtracking the increase in the sample size until we guarantee that $\mathbf{w}_n$ minimizes the empirical risk $R_n(\mathbf{w}_n)$ to within its statistical accuracy. This backtracking of the sample size is implemented in Step 11 and the optimality condition of $\mathbf{w}_n$ is checked in Step 12. The condition in Step 12 is on the gradient norm that, because $R_n$ is strongly convex, can be used to bound the suboptimality $R_n(\mathbf{w}_n) - R_n(\mathbf{w}_n^*)$ as

$$R_n(\mathbf{w}_n) - R_n(\mathbf{w}_n^*) \leq \frac{1}{2cV_n}\|\nabla R_n(\mathbf{w}_n)\|^2. \tag{14}$$

Observe that checking this condition requires an extra gradient computation undertaken in Step 10. That computation can be reused in the computation of the gradient in Step 5 once we exit the backtracking loop. We emphasize that when the condition in (13) is satisfied, there exists a sufficiently large $m$ for which the conditions in Theorem 1 are satisfied for $n = \alpha m$. This means that the backtracking condition in Step 12 is satisfied after one iteration and that, eventually, Ada Newton progresses by increasing the sample size by a factor $\alpha$. This means that Algorithm 1 can be thought of as having a damped phase where the sample size increases by a factor smaller than $\rho$ and a geometric phase where the sample size grows by a factor $\rho$ in all subsequent iterations. The computational cost of this geometric phase is of not more than $\alpha/(\alpha - 1)$ passes over the dataset and requires inverting not more than $\log_\alpha N$ Hessians. If $c > 16(2\sqrt{M} + 1)^2$, we make $\alpha = 2$ for optimizing to within statistical accuracy in about 2 passes over the dataset and after inversion of about $3.32 \log_{10} N$ Hessians.

## 4 Convergence Analysis

In this section we study the proof of Theorem 1. The main idea of the Ada Newton algorithm is introducing a policy for increasing the size of training set from $m$ to $n$ in a way that the current variable $\mathbf{w}_m$ is in the Newton quadratic convergence phase for the next regularized empirical risk $R_n$. In the following proposition, we characterize the required condition to guarantee staying in the local neighborhood of Newton's method.

**Proposition 3.** *Consider the sets $\mathcal{S}_m$ and $\mathcal{S}_n$ as subsets of the training set $\mathcal{T}$ such that $\mathcal{S}_m \subset \mathcal{S}_n \subset \mathcal{T}$. We assume that the number of samples in the sets $\mathcal{S}_m$ and $\mathcal{S}_n$ are $m$ and $n$, respectively. Further, define $\mathbf{w}_m$ as an $V_m$ optimal solution of the risk $R_m$, i.e., $R_m(\mathbf{w}_m) - R_m(\mathbf{w}_m^*) \leq V_m$. In addition, define $\lambda_n(\mathbf{w}) := \left(\nabla R_n(\mathbf{w})^T \nabla^2 R_n(\mathbf{w})^{-1} \nabla R_n(\mathbf{w})\right)^{1/2}$ as the Newton decrement of variable $\mathbf{w}$*

*associated with the risk $R_n$. If Assumption 1-3 hold, then Newton's method at point $\mathbf{w}_m$ is in the quadratic convergence phase for the objective function $R_n$, i.e., $\lambda_n(\mathbf{w}_m) < 1/4$, if we have*

$$\left(\frac{2(M + cV_m)V_m}{cV_n}\right)^{1/2} + \frac{(2(n-m)/n)V_n^{1/2} + (\sqrt{2c} + 2\sqrt{c} + c\|\mathbf{w}^*\|)(V_m - V_n)}{(cV_n)^{1/2}} \le \frac{1}{4} \quad w.h.p. \tag{15}$$

*Proof.* See Section 7.1 in the supplementary material. $\qquad\square$

From the analysis of Newton's method we know that if the Newton decrement $\lambda_n(\mathbf{w})$ is smaller than $1/4$, the variable $\mathbf{w}$ is in the local neighborhood of Newton's method; see e.g., Chapter 9 of [4]. From the result in Proposition 3, we obtain a sufficient condition to guarantee that $\lambda_n(\mathbf{w}_m) < 1/4$ which implies that $\mathbf{w}_m$, which is a $V_m$ optimal solution for the regularized empirical loss $R_m$, i.e., $R_m(\mathbf{w}_m) - R_m(\mathbf{w}_m^*) \le V_m$, is in the local neighborhood of the optimal argument of $R_n$ that Newton's method converges quadratically.

Unfortunately, the quadratic convergence of Newton's method for self-concordant functions is in terms of the Newton decrement $\lambda_n(\mathbf{w})$ and it does not necessary guarantee quadratic convergence in terms of objective function error. To be more precise, we can show that $\lambda_n(\mathbf{w}_n) \le \gamma\lambda_n(\mathbf{w}_m)^2$; however, we can not conclude that the quadratic convergence of Newton's method implies $R_n(\mathbf{w}_n) - R_n(\mathbf{w}_n^*) \le \gamma'(R_n(\mathbf{w}_m) - R_n(\mathbf{w}_n^*))^2$. In the following proposition we try to characterize an upper bound for the error $R_n(\mathbf{w}_n) - R_n(\mathbf{w}_n^*)$ in terms of the squared error $(R_n(\mathbf{w}_m) - R_n(\mathbf{w}_n^*))^2$ using the quadratic convergence property of Newton decrement.

**Proposition 4.** *Consider $\mathbf{w}_m$ as a variable that is in the local neighborhood of the optimal argument of the risk $R_n$ where Newton's method has a quadratic convergence rate, i.e., $\lambda_n(\mathbf{w}_m) \le 1/4$. Recall the definition of the variable $\mathbf{w}_n$ in (7) as the updated variable using Newton step. If Assumption 1 and 2 hold, then the difference $R_n(\mathbf{w}_n) - R_n(\mathbf{w}_n^*)$ is upper bounded by*

$$R_n(\mathbf{w}_n) - R_n(\mathbf{w}_n^*) \le 144(R_n(\mathbf{w}_m) - R_n(\mathbf{w}_n^*))^2. \tag{16}$$

*Proof.* See Section 7.2 in the supplementary material. $\qquad\square$

The result in Proposition 4 provides an upper bound for the sub-optimality $R_n(\mathbf{w}_n) - R_n(\mathbf{w}_n^*)$ in terms of the sub-optimality of variable $\mathbf{w}_m$ for the risk $R_n$, i.e., $R_n(\mathbf{w}_m) - R_n(\mathbf{w}_n^*)$. Recall that we know that $\mathbf{w}_m$ is in the statistical accuracy of $R_m$, i.e., $R_m(\mathbf{w}_m) - R_m(\mathbf{w}_m^*) \le V_m$, and we aim to show that the updated variable $\mathbf{w}_n$ stays in the statistical accuracy of $R_n$, i.e., $R_n(\mathbf{w}_n) - R_n(\mathbf{w}_n^*) \le V_n$. This can be done by showing that the upper bound for $R_n(\mathbf{w}_n) - R_n(\mathbf{w}_n^*)$ in (16) is smaller than $V_n$. We proceed to derive an upper bound for the sub-optimality $R_n(\mathbf{w}_m) - R_n(\mathbf{w}_n^*)$ in the following proposition.

**Proposition 5.** *Consider the sets $\mathcal{S}_m$ and $\mathcal{S}_n$ as subsets of the training set $\mathcal{T}$ such that $\mathcal{S}_m \subset \mathcal{S}_n \subset \mathcal{T}$. We assume that the number of samples in the sets $\mathcal{S}_m$ and $\mathcal{S}_n$ are $m$ and $n$, respectively. Further, define $\mathbf{w}_m$ as an $V_m$ optimal solution of the risk $R_m$, i.e., $R_m(\mathbf{w}_m) - R_m^* \le V_m$. If Assumption 1-3 hold, then the empirical risk error $R_n(\mathbf{w}_m) - R_n(\mathbf{w}_n^*)$ of the variable $\mathbf{w}_m$ corresponding to the set $\mathcal{S}_n$ is bounded above by*

$$R_n(\mathbf{w}_m) - R_n(\mathbf{w}_n^*) \le V_m + \frac{2(n-m)}{n}\left(V_{n-m} + V_m\right) + 2\left(V_m - V_n\right) + \frac{c(V_m - V_n)}{2}\|\mathbf{w}^*\|^2 \quad w.h.p. \tag{17}$$

*Proof.* See Section 7.3 in the supplementary material. $\qquad\square$

The result in Proposition 5 characterizes the sub-optimality of the variable $\mathbf{w}_m$, which is an $V_m$ sub-optimal solution for the risk $R_m$, with respect to the empirical risk $R_n$ associated with the set $\mathcal{S}_n$.

The results in Proposition 3, 4, and 5 lead to the result in Theorem 1. To be more precise, from the result in Proposition 3 we obtain that the condition in (10) implies that $\mathbf{w}_m$ is in the local neighborhood of the optimal argument of $R_n$ and $\lambda_n(\mathbf{w}_m) \le 1/4$. Hence, the hypothesis of Proposition 4 is satisfied and we have $R_n(\mathbf{w}_n) - R_n(\mathbf{w}_n^*) \le 144(R_n(\mathbf{w}_m) - R_n(\mathbf{w}_n^*))^2$. This result paired with the result in Proposition 5 shows that if the condition in (11) is satisfied we can conclude that $R_n(\mathbf{w}_n) - R_n(\mathbf{w}_n^*) \le V_n$ which completes the proof of Theorem 1.

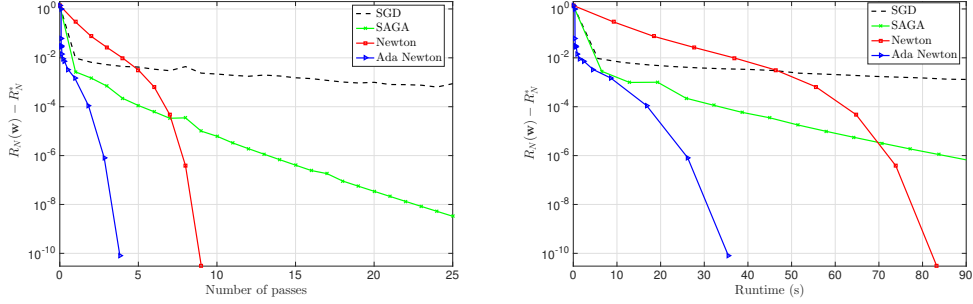

Figure 1: Comparison of SGD, SAGA, Newton, and Ada Newton in terms of number of effective passes over dataset (left) and runtime (right) for the protein homology dataset.

## 5  Experiments

In this section, we study the performance of Ada Newton and compare it with state-of-the-art in solving a large-scale classification problem. In the main paper we only use the protein homology dataset provided on KDD cup 2004 website. Further numerical experiments on various datasets can be found in Section 7.4 in the supplementary material. The protein homology dataset contains $N = 145751$ samples and the dimension of each sample is $p = 74$. We consider three algorithms to compare with the proposed Ada Newton method. One of them is the classic Newton's method with backtracking line search. The second algorithm is Stochastic Gradient Descent (SGD) and the last one is the SAGA method introduced in [6]. In our experiments, we use logistic loss and set the regularization parameters as $c = 200$ and $V_n = 1/n$.

The stepsize of SGD in our experiments is $2 \times 10^{-2}$. Note that picking larger stepsize leads to faster but less accurate convergence and choosing smaller stepsize improves the accuracy convergence with the price of slower convergence rate. The stepsize for SAGA is hand-optimized and the best performance has been observed for $\alpha = 0.2$ which is the one that we use in the experiments. For Newton's method, the backtracking line search parameters are $\alpha = 0.4$ and $\beta = 0.5$. In the implementation of Ada Newton we increase the size of the training set by factor 2 at each iteration, i.e., $\alpha = 2$ and we observe that the condition $\|\nabla R_n(\mathbf{w}_n)\| < (\sqrt{2c})V_n$ is always satisfied and there is no need for reducing the factor $\alpha$. Moreover, the size of initial training set is $m_0 = 124$. For the warmup step that we need to get into to the quadratic neighborhood of Newton's method we use the gradient descent method. In particular, we run gradient descent with stepsize $10^{-3}$ for 100 iterations. Note that since the number of samples is very small at the beginning, $m_0 = 124$, and the regularizer is very large, the condition number of problem is very small. Thus, gradient descent is able to converge to a good neighborhood of the optimal solution in a reasonable time. Notice that the computation of this warm up process is very low and is equal to 12400 gradient evaluations. This number of samples is less than 10% of the full training set. In other words, the cost is less than 10% of one pass over the dataset. Although, this cost is negligible, we consider it in comparison with SGD, SAGA, and Newton's method. We would like to mention that other algorithms such as Newton's method and stochastic algorithms can also be used for the warm up process; however, the gradient descent method sounds the best option since the gradient evaluation is not costly and the problem is well-conditioned for a small training set .

The left plot in Figure 1 illustrates the convergence path of SGD, SAGA, Newton, and Ada Newton for the protein homology dataset. Note that the $x$ axis is the total number of samples used divided by the size of the training set $N = 145751$ which we call number of passes over the dataset. As we observe, The best performance among the four algorithms belongs to Ada Newton. In particular, Ada Newton is able to achieve the accuracy of $R_N(\mathbf{w}) - R_N^* < 1/N$ by 2.4 passes over the dataset which is very close to theoretical result in Theorem 1 that guarantees accuracy of order $O(1/N)$ after $\alpha/(\alpha - 1) = 2$ passes over the dataset. To achieve the same accuracy of $1/N$ Newton's method requires 7.5 passes over the dataset, while SAGA needs 10 passes. The SGD algorithm can not achieve the statistical accuracy of order $O(1/N)$ even after 25 passes over the dataset.

Although, Ada Newton and Newton outperform SAGA and SGD, their computational complexity are different. We address this concern by comparing the algorithms in terms of runtime. The right

plot in Figure 1 demonstrates the convergence paths of the considered methods in terms of runtime. As we observe, Newton's method requires more time to achieve the statistical accuracy of $1/N$ relative to SAGA. This observation justifies the belief that Newton's method is not practical for large-scale optimization problems, since by enlarging $p$ or making the initial solution worse the performance of Newton's method will be even worse than the ones in Figure 1. Ada Newton resolves this issue by starting from small sample size which is computationally less costly. Ada Newton also requires Hessian inverse evaluations, but the number of inversions is proportional to $\log_\alpha N$. Moreover, the performance of Ada Newton doesn't depend on the initial point and the warm up process is not costly as we described before. We observe that Ada Newton outperforms SAGA significantly. In particular it achieves the statistical accuracy of $1/N$ in less than 25 seconds, while SAGA achieves the same accuracy in 62 seconds. Note that since the variable $\mathbf{w}_N$ is in the quadratic neighborhood of Newton's method for $R_N$ the convergence path of Ada Newton becomes quadratic eventually when the size of the training set becomes equal to the size of the full dataset. It follows that the advantage of Ada Newton with respect to SAGA is more significant if we look for a sub-optimality less than $V_n$. We have observed similar performances for other datasets such as A9A, W8A, COVTYPE, and SUSY – see Section 7.4 in the supplementary material.

## 6   Discussions

As explained in Section 4, Theorem 1 holds because condition (10) makes $\mathbf{w}_m$ part of the quadratic convergence region of $R_n$. From this fact, it follows that the Newton iteration makes the suboptimality gap $R_n(\mathbf{w}_n) - R_n(\mathbf{w}_n^*)$ the square of the suboptimality gap $R_n(\mathbf{w}_m) - R_n(\mathbf{w}_n^*)$. This yields condition (11) and is the fact that makes Newton steps valuable in increasing the sample size. If we replace Newton iterations by any method with linear convergence rate, the orders of both sides on condition (11) are the same. This would make aggressive increase of the sample size unlikely.

In Section 1 we pointed out four reasons that challenge the development of stochastic Newton methods. It would not be entirely accurate to call Ada Newton a stochastic method because it doesn't rely on stochastic descent directions. It is, nonetheless, a method for ERM that makes pithy use of the dataset. The challenges listed in Section 1 are overcome by Ada Newton because:

  **(i)** Ada Newton does not use line searches. Optimality improvement is guaranteed by increasing the sample size.

  **(ii)** The advantages of Newton's method are exploited by increasing the sample size at a rate that keeps the solution for sample size $m$ in the quadratic convergence region of the risk associated with sample size $n = \alpha m$. This allows aggressive growth of the sample size.

  **(iii)** The ERM problem is not necessarily strongly convex. A regularization of order $V_n$ is added to construct the empirical risk $R_n$

  **(iv)** Ada Newton inverts approximately $\log_\alpha N$ Hessians. To be more precise, the total number of inversion could be larger than $\log_\alpha N$ because of the backtracking step. However, the backtracking step is bypassed when the number of samples is sufficiently large.

It is fair to point out that items (ii) and (iv) are true only to the extent that the damped phase in Algorithm 1 is not significant. Our numerical experiments indicate that this is true but the conclusion is not warranted by out theoretical bounds except when the dataset is very large. This suggests the bounds are loose and that further research is warranted to develop tighter bounds.

## Footnotes

[1] [*]The first two authors have contributed equally in this work.

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
