[Supplementary Material · Supplementary_Material.pdf]

# 7 Supplementary Material

In this section we study the proofs of Propositions 3-5 and provide additional numerical experiments for the proposed Ada Newton method. To do so, first we prove Lemmata 6 and 7 which are intermediate results that we use in proving the mentioned propositions.

We start the analysis by providing an upper bound for the difference between the loss functions $L_n$ and $L_m$. The upper bound is studied in the following lemma which uses the condition in (3).

**Lemma 6.** *Consider $L_n$ and $L_m$ as the empirical losses of the sets $\mathcal{S}_n$ and $\mathcal{S}_m$, respectively, where they are chosen such that $\mathcal{S}_m \subset \mathcal{S}_n$. If we define $n$ and $m$ as the number of samples in the training sets $\mathcal{S}_n$ and $\mathcal{S}_m$, respectively, then the absolute value of the difference between the empirical losses is bounded above by*

$$|L_n(\mathbf{w}) - L_m(\mathbf{w})| \leq \frac{n-m}{n} \left( V_{n-m} + V_m \right), \qquad w.h.p. \tag{18}$$

*for any $\mathbf{w}$.*

*Proof.* First we characterize the difference between the difference of the loss functions associated with the sets $\mathcal{S}_m$ and $\mathcal{S}_n$. To do so, consider the difference

$$L_n(\mathbf{w}) - L_m(\mathbf{w}) = \frac{1}{n} \sum_{i \in \mathcal{S}_n} f_i(\mathbf{w}) - \frac{1}{m} \sum_{i \in \mathcal{S}_m} f_i(\mathbf{w}). \tag{19}$$

Notice that the set $\mathcal{S}_m$ is a subset of the set $\mathcal{S}_n$ and we can write $\mathcal{S}_n = \mathcal{S}_m \cup \mathcal{S}_{n-m}$. Thus, we can rewrite the right hand side of (19) as

$$L_n(\mathbf{w}) - L_m(\mathbf{w}) = \frac{1}{n} \left[ \sum_{i \in \mathcal{S}_m} f_i(\mathbf{w}) + \sum_{i \in \mathcal{S}_{n-m}} f_i(\mathbf{w}) \right] - \frac{1}{m} \sum_{i \in \mathcal{S}_m} f_i(\mathbf{w})$$

$$= \frac{1}{n} \sum_{i \in \mathcal{S}_{n-m}} f_i(\mathbf{w}) - \frac{n-m}{mn} \sum_{i \in \mathcal{S}_m} f_i(\mathbf{w}). \tag{20}$$

Factoring $(n-m)/n$ from the terms in the right hand side of (20) follows

$$L_n(\mathbf{w}) - L_m(\mathbf{w}) = \frac{n-m}{n} \left[ \frac{1}{n-m} \sum_{i \in \mathcal{S}_{n-m}} f_i(\mathbf{w}) - \frac{1}{m} \sum_{i \in \mathcal{S}_m} f_i(\mathbf{w}) \right]. \tag{21}$$

Now add and subtract the statistical loss $L(\mathbf{w})$ to obtain

$$|L_n(\mathbf{w}) - L_m(\mathbf{w})| = \frac{n-m}{n} \left| \frac{1}{n-m} \sum_{i \in \mathcal{S}_{n-m}} f_i(\mathbf{w}) - L(\mathbf{w}) + L(\mathbf{w}) - \frac{1}{m} \sum_{i \in \mathcal{S}_m} f_i(\mathbf{w}) \right|$$

$$\leq \frac{n-m}{n} \left( V_{n-m} + V_m \right), \tag{22}$$

where the last inequality follows by using the triangle inequality and the upper bound in (3). $\qquad\square$

The result in Lemma 6 shows that the upper bound for the difference between the loss functions associated with the sets $\mathcal{S}_m$ and $\mathcal{S}_n$ where $\mathcal{S}_m \subset \mathcal{S}_n$ is proportional to the difference between the size of these two sets $n-m$. This result will help us later to understand how much we can increase the size of the training set at each iteration. In other words, how large the difference $n-m$ could be, while we have the statistical accuracy.

In the following lemma, we characterize an upper bound for the norm of the optimal argument $\mathbf{w}_n^*$ of the empirical risk $R_n(\mathbf{w})$ in terms of the norm of statistical average loss $L(\mathbf{w})$ optimal argument $\mathbf{w}^*$.

**Lemma 7.** *Consider $L_n$ as the empirical loss of the set $\mathcal{S}_n$ and $L$ as the statistical average loss. Moreover, recall $\mathbf{w}^*$ as the optimal argument of the statistical average loss $L$, i.e., $\mathbf{w}^* = argmin_{\mathbf{w}} L(\mathbf{w})$. If Assumption 1 holds, then the norm of the optimal argument $\mathbf{w}_n^*$ of the regularized empirical risk $R_n(\mathbf{w}) := L_n(\mathbf{w}) + cV_n\|\mathbf{w}\|^2$ is bounded above by*

$$\|\mathbf{w}_n^*\|^2 \leq \frac{4}{c} + \|\mathbf{w}^*\|^2, \qquad w.h.p. \tag{23}$$

*Proof.* The optimality condition of $\mathbf{w}_n^*$ for the the regularized empirical risk $R_n(\mathbf{w}) = L_n(\mathbf{w}) + (cV_n)/2\|\mathbf{w}\|^2$ implies that

$$L_n(\mathbf{w}_n^*) + \frac{cV_n}{2}\|\mathbf{w}_n^*\|^2 \leq L_n(\mathbf{w}^*) + \frac{cV_n}{2}\|\mathbf{w}^*\|^2. \tag{24}$$

By regrouping the terms we can show that the squared norm $\|\mathbf{w}_n^*\|^2$ is bonded above by

$$\|\mathbf{w}_n^*\|^2 \leq \frac{2}{cV_n}\left(L_n(\mathbf{w}^*) - L_n(\mathbf{w}_n^*)\right) + \|\mathbf{w}^*\|^2. \tag{25}$$

We proceed to bound the difference $L_n(\mathbf{w}^*) - L_n(\mathbf{w}_n^*)$. By adding and subtracting the terms $L(\mathbf{w}^*)$ and $L(\mathbf{w}_n^*)$ we obtain that

$$L_n(\mathbf{w}^*) - L_n(\mathbf{w}_n^*) = \left[L_n(\mathbf{w}^*) - L(\mathbf{w}^*)\right] + \left[L(\mathbf{w}^*) - L(\mathbf{w}_n^*)\right] + \left[L(\mathbf{w}_n^*) - L_n(\mathbf{w}_n^*)\right]. \tag{26}$$

Notice that the second bracket in (26) is non-positive since $L(\mathbf{w}^*) \leq L(\mathbf{w}_n^*)$. Therefore, it is bounded by 0. According to (3), the first and third brackets in (26) are with high probability bounded above by $V_n$. Replacing these upper bounds by the brackets in (26) yields

$$L_n(\mathbf{w}^*) - L_n(\mathbf{w}_n^*) \leq 2V_n. \tag{27}$$

Substituting the upper bound in (27) into (25) implies the claim in (23). $\qquad\square$

## 7.1 Proof of Proposition 3

From the self-concordance analysis of Newton's method we know that the variable $\mathbf{w}_m$ is in the neighborhood that Newton's method has a quadratic convergence rate if $\lambda_n(\mathbf{w}_m) \leq 1/4$; see e.g., Chapter 9 of [4]. We proceed to come up with a condition for the quadratic convergence phase which guarantees that $\lambda_n(\mathbf{w}_m) < 1/4$ and $\mathbf{w}_m$ is in the local neighborhood of the optimal argument of $R_n$. Recall that we have a $\mathbf{w}_m$ which has sub-optimality $V_m$ for $R_m$. We then proceed to enlarge the sample size to $n$ and start from the observation that we can bound $\lambda_n(\mathbf{w}_m)$ as

$$\lambda_n(\mathbf{w}_m) = \|\nabla R_n(\mathbf{w}_m)\|_{\mathbf{H}_n^{-1}} \leq \|\nabla R_m(\mathbf{w}_m)\|_{\mathbf{H}_n^{-1}} + \|\nabla R_n(\mathbf{w}_m) - \nabla R_m(\mathbf{w}_m)\|_{\mathbf{H}_n^{-1}}, \tag{28}$$

where we have used the definition $\mathbf{H}_n = \nabla^2 R_n(\mathbf{w}_m)$. Note that the weighted norm $\|\mathbf{a}\|_{\mathbf{A}}$ for vector $\mathbf{a}$ and matrix $\mathbf{A}$ is equal to $\|\mathbf{a}\|_{\mathbf{A}} = (\mathbf{a}^T \mathbf{A} \mathbf{a})^{1/2}$. First, we bound the norm $\|\nabla R_n(\mathbf{w}_m)\|_{\mathbf{H}_n^{-1}}$ in (28). Notice that the Hessian $\nabla^2 R_n(\mathbf{w}_m)$ can be written as $\nabla^2 L_n(\mathbf{w}_m) + cV_n\mathbf{I}$. Thus, the eigenvalues of the Hessian $\mathbf{H}_n = \nabla^2 R_n(\mathbf{w}_m)$ are bounded below by $cV_n$ and consequently the eigenvalues of the Hessian inverse $\mathbf{H}_n^{-1} = \nabla^2 R_n(\mathbf{w}_m)^{-1}$ are upper bounded by $1/(cV_n)$. This bound implies that $\|\mathbf{H}_n^{-1}\| \leq 1/(cV_n)$. Moreover, from Theorem 2.1.5 of [15], we know that the Lipschitz continuity of the gradients $\nabla R_m(\mathbf{w})$ with constant $M + cV_m$ implies that

$$\|\nabla R_m(\mathbf{w}_m)\|^2 \leq 2(M + cV_m)(R_m(\mathbf{w}_m) - R_m(\mathbf{w}_m^*)) \leq 2(M + cV_m)V_m, \tag{29}$$

where the last inequality holds comes from the condition that $R_m(\mathbf{w}_m) - R_m(\mathbf{w}_m^*) \leq V_m$. Considering the upper bound for $\|\nabla R_m(\mathbf{w}_m)\|^2$ in (29) and the inequality $\|\nabla^2 R_n(\mathbf{w}_m)^{-1}\| \leq 1/(cV_n)$ we can write

$$\|\nabla R_m(\mathbf{w}_m)\|_{\mathbf{H}_n^{-1}} = \left[\nabla R_m(\mathbf{w}_m)^T \mathbf{H}_n^{-1} \nabla R_m(\mathbf{w}_m)\right]^{1/2} \leq \left(\frac{2(M + cV_m)V_m}{cV_n}\right)^{1/2}. \tag{30}$$

Now we proceed to bound the second the term in (28). The definition of the risk function the gradient can be written as $\nabla R_n(\mathbf{w}) = \nabla L_n(\mathbf{w}) + (cV_n)\mathbf{w}$. Thus, we can derive an upper bound for the difference $\|\nabla R_n(\mathbf{w}_m) - \nabla R_m(\mathbf{w}_m)\|$ as

$$\begin{aligned}
&\|\nabla R_n(\mathbf{w}_m) - \nabla R_m(\mathbf{w}_m)\| \\
&\qquad \leq \|\nabla L_n(\mathbf{w}_m) - \nabla L_m(\mathbf{w}_m)\| + c(V_m - V_n)\|\mathbf{w}_m\| \\
&\qquad \leq \|\nabla L_n(\mathbf{w}_m) - \nabla L_m(\mathbf{w}_m)\| + c(V_m - V_n)\|\mathbf{w}_m - \mathbf{w}_m^*\| + c(V_m - V_n)\|\mathbf{w}_m^*\|,
\end{aligned} \tag{31}$$

where in the second inequality we have used the triangle inequality and replaced $\|\mathbf{w}_m\|$ by its upper bound $\|\mathbf{w}_m - \mathbf{w}_m^*\| + \|\mathbf{w}_m^*\|$. By following the steps in (19)-(22) we can show that the difference $\|\nabla L_n(\mathbf{w}_m) - \nabla L_m(\mathbf{w}_m)\|$ is bounded above by

$$\begin{aligned}
\|\nabla L_n(\mathbf{w}) - \nabla L_m(\mathbf{w})\| &\leq \frac{n-m}{n}\|\nabla L_{n-m}(\mathbf{w}) - \nabla L(\mathbf{w})\| + \frac{n-m}{n}\|\nabla L_m(\mathbf{w}) - \nabla L(\mathbf{w})\| \\
&\leq \frac{2(n-m)}{n}V_n^{1/2},
\end{aligned} \tag{32}$$

where the second inequality uses the condition that $\|\nabla L_m(\mathbf{w}) - \nabla L(\mathbf{w})\| \leq V_m^{1/2}$ as in Assumption 3.

Note that the strong convexity of the risk $R_m$ with parameter $cV_m$ yields

$$\|\mathbf{w}_m - \mathbf{w}_m^*\|^2 \leq \frac{2}{cV_m}(R_m(\mathbf{w}_m) - R_m(\mathbf{w}_m^*)) \leq \frac{2}{c}. \tag{33}$$

Thus, by considering the inequalities in (32) and (33) we can show that upper bound in (31) can be replaced by

$$\|\nabla R_n(\mathbf{w}_m) - \nabla R_m(\mathbf{w}_m)\| \leq \frac{2(n-m)}{n}V_n^{1/2} + (\sqrt{2c} + c\|\mathbf{w}_m^*\|)(V_m - V_n). \tag{34}$$

Substituting the upper bounds in (30) and (34) for the first and second summands in (28), respectively, follows the inequality

$$\lambda_n(\mathbf{w}_m) \leq \left(\frac{2(M + cV_m)V_m}{cV_n}\right)^{1/2} + \frac{(2(n-m)/n)V_n^{1/2} + (\sqrt{2c} + c\|\mathbf{w}_m^*\|)(V_m - V_n)}{(cV_n)^{1/2}}. \tag{35}$$

Note that the result in (23) shows that $\|\mathbf{w}_m^*\|^2 \leq (4/c) + \|\mathbf{w}^*\|^2$ with high probability. This observation implies that $\|\mathbf{w}_m^*\|$ is bounded above by $(2/\sqrt{c}) + \|\mathbf{w}^*\|$. Replacing the norm $\|\mathbf{w}_m^*\|$ in (35) by the upper bound $(2/\sqrt{c}) + \|\mathbf{w}^*\|$ yields

$$\lambda_n(\mathbf{w}_m) \leq \left(\frac{2(M + cV_m)V_m}{cV_n}\right)^{1/2} + \frac{(2(n-m)/n)V_n^{1/2} + (\sqrt{2c} + 2\sqrt{c} + c\|\mathbf{w}^*\|)(V_m - V_n)}{(cV_n)^{1/2}}. \tag{36}$$

As we mentioned previously, the variable $\mathbf{w}_m$ is in the neighborhood that Newton's method has a quadratic convergence rate for the function $R_n$ if the condition $\lambda_n(\mathbf{w}_m) \leq 1/4$ holds. Hence, if the right hand side of (36) is bounded above by $1/4$ we can conclude that $\mathbf{w}_m$ is in the local neighborhood and the proof is complete.

## 7.2 Proof of Proposition 4

To prove the result in (16) first we need to find upper and lower bounds for the difference $R_n(\mathbf{w}) - R_n(\mathbf{w}_n^*)$ in terms of the Newton decrement parameter $\lambda_n(\mathbf{w})$. To do so, we use the result in Theorem 4.1.11 of [15] which shows that

$$\lambda_n(\mathbf{w}) - \ln(1 + \lambda_n(\mathbf{w})) \leq R_n(\mathbf{w}) - R_n(\mathbf{w}_n^*) \leq -\lambda_n(\mathbf{w}) - \ln(1 - \lambda_n(\mathbf{w})). \tag{37}$$

Note that we assume that $0 < \lambda_n(\mathbf{w}) < 1/4$. Thus, we can use the Taylor's expansion of $\ln(1 + a)$ for $a = \lambda_n(\mathbf{w})$ to show that $\lambda_n(\mathbf{w}) - \ln(1 + \lambda_n(\mathbf{w}))$ is bounded below by $(1/2)\lambda_n(\mathbf{w})^2 - (1/3)\lambda_n(\mathbf{w})^3$. Since $0 < \lambda_n(\mathbf{w}) < 1/4$ we can show that $(1/6)\lambda_n(\mathbf{w})^2 \leq (1/2)\lambda_n(\mathbf{w})^2 - (1/3)\lambda_n(\mathbf{w})^3$. Thus, the term $\lambda_n(\mathbf{w}) - \ln(1 + \lambda_n(\mathbf{w}))$ is bounded below by $(1/6)\lambda^2$. Likewise, we use Taylor's expansion of $\ln(1 - a)$ for $a = \lambda_n(\mathbf{w})$ to show that $-\lambda_n(\mathbf{w}) - \ln(1 - \lambda_n(\mathbf{w}))$ is bounded above by $\lambda_n(\mathbf{w})^2$ for $\lambda_n(\mathbf{w}) < 1/4$; see e.g., Chapter 9 of [4]. Considering these bounds and the inequalities in (37) we can write

$$\frac{1}{6}\lambda_n(\mathbf{w})^2 \leq R_n(\mathbf{w}) - R_n(\mathbf{w}_n^*) \leq \lambda_n(\mathbf{w})^2. \tag{38}$$

Recall that the variable $\mathbf{w}_m$ satisfies the condition $\lambda_n(\mathbf{w}_m) \leq 1/4$. Thus, according to the quadratic convergence rate of Newton's method for self-concordant functions [4], we know that the Newton decrement has a quadratic convergence and we can write

$$\lambda_n(\mathbf{w}_n) \leq 2\lambda_n(\mathbf{w}_m)^2. \tag{39}$$

We use the result in (38) and (39) to show that the optimality error $R_n(\mathbf{w}_n) - R_n(\mathbf{w}_n^*)$ has an upper bound which is proportional to $(R_n(w_m) - R_n(\mathbf{w}_n^*))^2$. In particular, we can write $R_n(\mathbf{w}_n) - R_n(\mathbf{w}_n^*) \leq \lambda_n(\mathbf{w}_n)^2$ based on the second inequality in (38). This observation in conjunction with the result in (39) implies that

$$R_n(\mathbf{w}_n) - R_n(\mathbf{w}_n^*) \leq 4\lambda_n(\mathbf{w}_m)^4. \tag{40}$$

The first inequality in (38) implies that $\lambda_n(\mathbf{w}_m)^4 \leq 36(R_n(\mathbf{w}_m) - R_n(\mathbf{w}_n^*))^2$. Thus, we can substitute $\lambda_n(\mathbf{w}_m)^4$ in (40) by $36(R_n(\mathbf{w}_m) - R_n(\mathbf{w}_n^*))^2$ to obtain the result in (16).

## 7.3 Proof of Proposition 5

Note that the difference $R_n(\mathbf{w}_m) - R_n(\mathbf{w}_n^*)$ can be written as

$$R_n(\mathbf{w}_m) - R_n(\mathbf{w}_n^*) = R_n(\mathbf{w}_m) - R_m(\mathbf{w}_m) + R_m(\mathbf{w}_m) - R_m(\mathbf{w}_m^*)$$
$$+ R_m(\mathbf{w}_m^*) - R_m(\mathbf{w}_n^*) + R_m(\mathbf{w}_n^*) - R_n(\mathbf{w}_n^*). \tag{41}$$

We proceed to bound the differences in (41). To do so, note that the difference $R_n(\mathbf{w}_m) - R_m(\mathbf{w}_m)$ can be simplified as

$$R_n(\mathbf{w}_m) - R_m(\mathbf{w}_m) = L_n(\mathbf{w}_m) - L_m(\mathbf{w}_m) + \frac{c(V_n - V_m)}{2}\|\mathbf{w}_m\|^2$$
$$\leq L_n(\mathbf{w}) - L_m(\mathbf{w}), \tag{42}$$

where the inequality follows from the fact that $V_n < V_m$ and $V_n - V_m$ is negative. It follows from the result in Lemma 6 that the right hand side of (42) is bounded by $(n-m)/n\,(V_{n-m} + V_m)$. Therefore,

$$R_n(\mathbf{w}_m) - R_m(\mathbf{w}_m) \leq \frac{n-m}{n}\,(V_{n-m} + V_m). \tag{43}$$

According to the fact that $\mathbf{w}_m$ as an $V_m$ optimal solution for the sub-optimality $R_m(\mathbf{w}_m) - R_m(\mathbf{w}_m^*)$ we know that

$$R_m(\mathbf{w}_m) - R_m(\mathbf{w}_m^*) \leq V_m. \tag{44}$$

Based on the definition of $\mathbf{w}_m^*$ which is the optimal solution of the risk $R_m$, the third difference in (41) which is $R_m(\mathbf{w}_m^*) - R_m(\mathbf{w}_n^*)$ is always negative. I.e.,

$$R_m(\mathbf{w}_m^*) - R_m(\mathbf{w}_n^*) \leq 0. \tag{45}$$

Moreover, we can use the triangle inequality to bound the difference $R_m(\mathbf{w}_n^*) - R_n(\mathbf{w}_n^*)$ in (41) as

$$R_m(\mathbf{w}_n^*) - R_n(\mathbf{w}_n^*) = L_m(\mathbf{w}_n^*) - L_n(\mathbf{w}_n^*) + \frac{c(V_m - V_n)}{2}\|\mathbf{w}_n^*\|^2$$
$$\leq \frac{n-m}{n}\,(V_{n-m} + V_m) + \frac{c(V_m - V_n)}{2}\|\mathbf{w}_n^*\|^2. \tag{46}$$

Replacing the differences in (41) by the upper bounds in (43)-(46) follows

$$R_n(\mathbf{w}_m) - R_n(\mathbf{w}_n^*) \leq V_m + \frac{2(n-m)}{n}\,(V_{n-m} + V_m) + \frac{c(V_m - V_n)}{2}\|\mathbf{w}_n^*\|^2 \quad \text{w.h.p.} \tag{47}$$

Substitute $\|\mathbf{w}_n^*\|^2$ in (47) by the upper bound in (23) to obtain the result in (17).

## 7.4 Additional Numerical Experiments

In this section, we compare the performance of SAGA, Newton, and Ada Newton in solving a $l_2$-regularized logistic regression on A9A, W8A, COVTYPE.BINARY, and SUSY datasets. These datasets have different size and dimensionality as stated in Table 1.

Table 1: Summary of the datasets

| Dataset | Number of Samples | Number of Features |
|---|---|---|
| A9A | 32561 | 123 |
| W8A | 49749 | 300 |
| COVTYPE.BINARY | 581012 | 54 |
| SUSY | 5000000 | 18 |

In these experiments, we use $90\%$ of samples of the data points as the training set and the remaining $10\%$ as the test set. The stepsize for SAGA is set as $1/L$ as suggested in [6].

Figure 2 illustrates the sub-optimality $R_N(\mathbf{w}) - R_N^*$ of these methods versus the number of passes over the datasets. In order to connect convergence on the empirical and expected risks, we plot the a

Figure 2: Comparison of the sub-optimality of SAGA, Newton, and Ada Newton in terms of number of effective passes over dataset for four datasets. The horizontal axis represents the number of effective passes over the training set and the vertical axis shows the sub-optimality error $R_N(\mathbf{w}) - R_N^*$ where $N$ is the size of training set. The dotted horizontal line refers to statistical accuracy.

horizontal dotted green line that shows the iteration at which Ada Newton reached convergence on the test set. As we observe, Ada Newton achieves statistical accuracy (the green line) after almost 2 passes over the training set for all the considered datasets.

Since the computational complexity of SAGA is lower than the ones for Newton's method and Ada Newton, we also compare these methods in terms of runtime. Figure 3 demonstrates the sub-optimality of these methods versus their runtimes. This comparison justifies that the Newton's method is impractical for large scale ERM minimization, and Ada Newton significantly improves the performance of Newton's method.

We further present the expected error of classifiers trained by SAGA, Newton, and Ada Newton on the test set of each of the considered datasets in Figure 4. The results showcase that in all experiments Ada Newton achieves a target test error faster than Newton's method and SAGA.

Figure 3: Comparison of the sub-optimality of SAGA, Newton, and Ada Newton in terms of run time for four datasets. The horizontal axis represents runtime and the vertical axis shows the sub-optimality error $R_N(\mathbf{w}) - R_N^*$ where $N$ is the size of training set. The dotted horizontal line refers to statistical accuracy.

Figure 4: Comparison of the sub-optimality of SAGA, Newton, and Ada Newton in terms of test error for four datasets. The horizontal axis represents the number of effective passes over the training set and the vertical axis shows the error on the test set.