[Reviews · NeurIPS 2016]

Reviewer 1

Summary

The authors study the ERM (Empirical Risk Minimization) problem and propose an Adaptive Newton algorithm for solving it to statistical accuracy. The idea is to consider a sequence of nested ERM problems of increasing sizes (each new ERM problem is described by all the examples of the previous ERM problem in the sequence and a number of fresh samples) up to a certain fixed size, and take a carefully designed Newton step (of unit stepsize) related to each ERM problem. The claim is that if this is done “right”, each ERM problem is solved up to statistical accuracy.

Qualitative Assessment

The paper contains a novel and interesting idea, and is well written. I think it should be accepted. This is one of a very few papers which attempt to combine optimization and statistical considerations. Most papers suggesting algorithms for solving the ERM problem simply focus on solving the deterministic ERM problem, and ignore the fact that ERM objective is an approximation to the true loss which arises as an expectation of the loss over an unknown sample distribution; and hence is subject to an approximation error. This is of course known in the literature, but it is notoriously difficult to address both the optimization aspect and the approximation aspect in a meaningful way in a single work. This papers takes a highly novel route to this problem; and I think there is a potential in this line of work. However, the paper is not flawless – below I will enumerate some of the issues with the paper. Some of these issues can be remedied by a revision, but some seem to be an intrinsic part of the proposed approach, and hard to get rid off. 1) The introduction enumerates 4 challenges left in the implementation of Newton-like methods for ERM. I agree that these issues remain with most of the approaches. However, there are approaches which circumvent these issues – but these are not mentioned by the authors. The paper would be stronger if a more balanced treatment was made. For instance, consider the SDNA algorithm of Qu, Richtarik, Takac and Fercoq (Proceedings of The 33rd International Conference on Machine Learning, pp. 1823–1832, 2016). For L2-reglarized ERM problems, this method takes Newton-like steps in random subspaces in the dual. i) The method has a global linear rate, better than SDCA, without any explicit line-search procedure. ii) Since there is no point in looking at local convergence beyond statistical accuracy, the analysis of SDNA focuses on speedup from mini-batching, which is more important. This speedup is superlinear in the mini-batch size, unlike all other stochastic first-order methods. iii) SDNA was analysed for strongly convex ERM problems – so this issue is present. iv) This issue is circumvented by inverting sufficiently small random principal submatrices of the Hessian in the dual (the Hessian does not change in the dual, which is the advantage of using the dual formulation). This is just one example of how *some* of these issues *are* addressed by existing research. Hence, the paper should be more precise in stating the current status of these challenges. 2) I do not see how the claim “Our theoretical and numerical analyses indicate that we 
can make alpha = 2. In that case, we can can optimize to within statistical accuracy in about 2 passes over 
the dataset and after inversion of about 3.32 log10 N Hessians. “ is supported in theory.
 Also, it is not clear enough what exactly the above statement means. It is made in the abstract, on lines 49-51 and lines 143-145. The precise complexity depends on values values of c and alpha, which in turn depend on the values of V_n, which are unknown in practice. The sizes of the constants in the O(1/n), O(1/sqrt{n}) decay rates will matter, and these will influence the precise constant elsewhere, including the constant 2 claimed above. So, I think this conclusion is exaggerated and needs moderation. 3) The same applies to the claim that the result is supported by experiments. Conclusions can’t be drawn from a single experiment using a single dataset (The sentence “We have observed similar performances for other datasets such as A9A, 
COVTYPE, and SUSY. 
” is much less satisfying than including the experiments). Another reason to moderate the statement and remove it from the abstract in order for the readers not to be potentially misled. 4) The authors should make a detailed complexity analysis of the method, including the cost of each iteration, and compare this to existing results. A the moment, no such comparison is given, and hence it is not straightforward how the results should be interpreted and why they should be appreciated. Are your complexity results better than those of existing methods, such as SAGA or SDCA or Quartz? 5) The algorithm seems to be conceptual in practice, and impractical (this is admitted on lines 126-127). This is because the method relies on the knowledge of parameters such as V_n for all n, but these are not known. Fine tuning these might be computationally difficult, and I would expect the actual solver to be slower than state of the art (SDCA, SAGA, Quartz, SAG, AdaSDCA, dfSDCA, SVRG, S2GD, Finito, …) on most datasets for this reason. There are more parameters which need to be tuned, such as alpha_0, beta, c. How would these be chosen in practice? Will the method be robust to this choice of parameters across different datasets? How will it compare to off-the shelf implementation of SDCA or S2GD, say, which do not require any parameters to be tuned? If the authors believe the method to be practical, which would enhance the paper considerably, then considerations of this type need to be included in the paper, and extensive computational experiments performed. 6) Theorem 1 (appearing in Section 3) relies on assumptions which are not stated until Section 4. The reader is not alerted to this. Likewise, Proposition 2 relies on these assumptions, but this is not explicitly stated. Moreover, the term M appears in the result there, unexpectedly, as this is the first time this term appears in the paper. I later found tat it was defined in a later section. Some adjustment is needed to the text in order to rectify the flow of ideas a bit and make the paper a bit easier to read in these places (the paper is very well written already; this is a minor issue which can be easily addressed in a number of ways). Small issues: 7) Line 50: cases -> case 8) Should not c be a parameter of Alg 1? 9) In several paces (e.g., lines 171, 189, …), you write “Assumption 1-3” instead of “Assumptions 1-3” 10) Line 178: that -> in which (might be best to break this sentence up into 2-3 shorter sentences) 11) Line 181: necessary -> necessarily 12) Line 264: The -> the Bibliography: There are many issues with the bibliography. Here are some I found: 1) [1]: capitalize journal name 2) [3]: capitalize Newton 3) [4]: include NIPS volume 4) [6]: appeared already in Proceedings of The 33rd International Conference on Machine Learning, pp. 1463–1471, 2016 5) [7]: Saga -> SAGA; also include NIPS volume 6) [10]: Bfgs -> BFGS; appeared: Proceedings of The 33rd International Conference on Machine Learning, 1869-1878, 2016 7) [11]: newton -> Newton 8) [12]: include NIPS volume 9) [13]: l-bfgs -> L-BFGS; appeared: Proceedings of the 19th International Conference on Artificial Intelligence and Statistics, pp. 249–258, 2016 10) [14] publisher?; volume I, not volume i 11) [15]: was published already (long time ago) 12) [18]: include NIPS volume 13) [19]: newton -> Newton 14) [23]: capitalize book title --- post rebuttal comments --- Thanks for the explanations; some of my concerns were addressed, but many were not (or not in the way I hoped). I still think highly of the paper (and think it should be accepted), but to be fair, I also believe the score I gave for clarity is higher than it really should be. I am decreasing the clarity score from 4 to 3.

Confidence in this Review

3-Expert (read the paper in detail, know the area, quite certain of my opinion)


Reviewer 2

Summary

The paper describes an approach to speeding up Newton's method for empirical risk minimization problems via subsampling. The approach stems from the observation that there is never any need to reduce empircal risk below statistical noise. The main result of the paper is to show that if we are given a sufficiently large sample and an iterate that has empirical risk on the sample that is below statistical noise, then only one Newton step (with a constant step length) is needed to reduce empirical risk below statistical noise on superset of the sample that is a constant factor larger (usually the factor is 2). This leads to the Ada Newton algorithm, which applies Newton steps to the empirical risk on increasingly larger samples of the entire dataset. Experimental results show that Ada Newton is faster than Newton's method and SAGA on a protein dataset.

Qualitative Assessment

I liked the paper. The theoretical results are strong, the experiments seem solid (though I wish they had been on more than one dataset), and the presentation is clear. My only issue is with a heuristic that is used by the Ada Newton algorithm. Before getting to that, I need a clarification: Has the inequality in Step 12 of the Ada Newton algorithm been inadvertently reversed? Because otherwise I don't understand how this can be an optimality condition. Assuming the inequality should be flipped, it seems to me that the backtracking phase could easily enter a cycle. For example, if \alpha = 2, \beta = 0.5, and the optimality condition is not satisfied after a single doubling of the sample size, then I think the algorithm will make no progress. Do the authors agree? If so, what can be done about this? Minor mistakes: - Line 71: w should be w_n. - Line 93: What are assumptions 1 - 3? This only becomes clear in Section 4. - Eq. 8 and 11: What is M? This only becomes clear in Section 4. - Line 135: Step 5 can't be the right step to reference here. == post-rebuttal answer== Reading the authors' rebuttal led me to spot another major typo in the paper: I think line 6 of Algorithm 1 should have a min, not a max (because the max is always N, right?).

Confidence in this Review

2-Confident (read it all; understood it all reasonably well)


Reviewer 3

Summary

In this paper, the authors proposed a new type of Newton’s method to solve the empirical risk minimization problem by using adaptive sample sizes. The author claimed that with proper parameters, only one iteration of Newton’s method is enough to reach the statistical accuracy based on the less accurate solution of the smaller-sized problem and thus the whole algorithm is much more efficient.

Qualitative Assessment

1. This paper proposed a new Newton-based method using adaptive sample sizes, which is novel in the field of Newton-based algorithms. It seems that the authors followed and further extended the paper Starting Small – Learning with Adaptive Sample Sizes from gradient method to Newton based method. 2. The author claimed that the proposed algorithm would need much less data passes in order to get desired result with proper conditions, specifically, only one Newton iteration is needed each time. However, since the algorithm also introduces backtracking mechanism, I assume that the conditions are not always hold and it means the algorithm would need more than one iteration. Thus it made me doubt that the algorithm would significantly improve the performance on data passes since the original Newton’s method has already enjoyed a fast convergence rate. 3. Comparing with theoretical results, the experiment part would be more convincing if adding dyna-SAGA method proposed in Starting Small – Learning with Adaptive Sample Sizes as a baseline method. 4. In proposition 2, equation 11, according to the proof behind it, the second term on the left hand side should contain a factor of (alpha-1)/alpha instead of alpha/(alpha -1) 5. The arrangement and the structure would be much clearer if the author could introduce the assumptions in Section 4 before Section 3 so that people wouldn’t get confused by the unintroduced parameters while reading Section 3. == post-rebuttal update== I have read the rebuttal

Confidence in this Review

2-Confident (read it all; understood it all reasonably well)


Reviewer 4

Summary

This paper proposes an adaptive setting so that at each iteration of the Newton method being applied to empirical risk minimization uses a subset of data. The way to adaptively decide Newton method is designed so that convergence results hold.

Qualitative Assessment

Overall I think the idea is interesting. However, a concern is that the current setting does not really solve empirical risk minimization for large-scale sets. The reason is that Hessian inverse is still needed. The algorithm becomes impractical if the number of features is large. In this regard, I think sub-Hessian method proposed by, for example, Byrd et al. (2011) are more practical.

Confidence in this Review

2-Confident (read it all; understood it all reasonably well)


Reviewer 5

Summary

This paper considers ERM for large datasets. A second order algorithm based on sampling data is proposed. The size of samples is based on the statistical accuracy needed during the iterative process. Convergence analysis and numerical results are provided.

Qualitative Assessment

This paper is well written, and the idea of combining L2 regularizer with desired statistical accuracy, V_n, is novel (to the best of my knowledge). However, I have concerns about the claims made by the authors in which they address the four current challenges of Newton method. The authors claim that the proposed algorithm does not need line searches and so it could save computational cost. The trade-off, however, is the sample size backtracking step in Algorithm 1. In each backtracking step, one needs to invert a system of linear equations which is in the same size of the problem, p. This is extremely expensive when p is reasonably large. Even though the sample size backtracking step is not needed once the algorithm enters its geometric phase, it does not outperform original Newton method, because Newton method can take stepsize 1 (without line search) in its quadratic phase. This concern of the cost of damped phase is also mentioned at the end of the paper. Also, in the numerical examples, the chosen datasets have small p's and so the general numerical performance of Ada Newton is not well analyzed. Also, I think it would be more fair to compare Ada Newton with sub-sample Newton in the numerical section [8]. I agree it is likely that Ada Newton would outperform original Newton method when p is small. However, there are some works focus this particular setting, see e.g. Newton-Stein Method: A Second Order Method for GLMs via Stein’s Lemma. Therefore, more justifications are needed in order to claim the contributions in this particular setting. == post-rebuttal answer== I agree with the authors that "Ada Newton is not the end of the road", and the proposed approach is worth to be investigated further. I look forward to the additional numerical results that the authors would provide.

Confidence in this Review

2-Confident (read it all; understood it all reasonably well)


Reviewer 6

Summary

This paper proposes a variant of Newton's method which defines a sequence of objective functions, where each objective can be solved to its statistical accuracy with a single Newton step (when initialized from a solution to the previous objective).

Qualitative Assessment

This is a nice paper. It is well motivated. And the contribution is original. Obtaining the benefits of second-order optimization algorithms in machine learning involves overcoming problems (i)-(iv) outlined by the authors in the paper. And progress in this area is valuable. Instead of applying Newton's method directly to a given objective, the authors instead define a sequence of (regularized) objectives each of which can be solved efficiently (to the relevant degree of precision) when initialized from the previous solution. There is a line of work on continuation methods for solving optimization problems that involve interpolating the objective from an easy one to the real one in a way that always leaves the solution within the basin of quadratic convergence for the current objective. Can you comment on how these approaches relate to your approach? I have not seem these applied to ERM but rather to linear programming, solving systems of (polynomial) equations, and things like that. I have some questions about the limitations of Ada Newton. In Figure 1, Ada Newton seems to be several times faster than regular Newton. But Newton's method is often plain inapplicable to many machine learning problems because of the high dimensionality of the optimization problem. In these cases, if Ada Newton is 10x faster, it will still not be applicable. What I'm asking is, are there situations where Ada Newton will actually be usable in practice, while Newton's method will be unusable? Or will it typically be several times faster?

Confidence in this Review

1-Less confident (might not have understood significant parts)